# Catalytic Transformation of Triglycerides to Biodiesel with SiO_2_-SO_3_H and Quaternary Ammonium Salts in Toluene or DMSO

**DOI:** 10.3390/molecules27030953

**Published:** 2022-01-30

**Authors:** Sandro L. Barbosa, Adeline C. Pereira Rocha, David Lee Nelson, Milton S. de Freitas, Antônio A. P. Fulgêncio Mestre, Stanlei I. Klein, Giuliano C. Clososki, Franco J. Caires, Danilo L. Flumignan, Letícia Karen dos Santos, Alexandre P. Wentz, Vânya M. Duarte Pasa, Regiane D. Fernandes Rios

**Affiliations:** 1Department of Pharmacy, Universidade Federal dos Vales do Jequitinhonha e Mucuri—UFVJM, R. da Glória, 187, Diamantina 39100-000, Brazil; adeline.rocha@ufvjm.edu.br (A.C.P.R.); dleenelson@gmail.com (D.L.N.); milton.freitas@ufvjm.edu.br (M.S.d.F.); antonio.alexandre@ufvjm.edu.br (A.A.P.F.M.); 2Department of General and Inorganic Chemistry, Institute of Chemistry, São Paulo State University—UNESP, R. Prof. Francisco Degni 55, Quitandinha, Araraquara 14800-900, Brazil; stanleiklein@gmail.com; 3Department of Physics and Chemistry, Faculdade de Ciências Farmacêuticas de Ribeirão Preto, São Paulo University—USP, Av. do Café s/n, Ribeirao Preto 14040-903, Brazil; gclososki@usp.br (G.C.C.); fjcaires@usp.br (F.J.C.); 4Instituto Federal de Educação, Ciência e Tecnologia de Mato Grosso—IFMT—Campus Cuiabá, Departamento das Áreas de Base Comum (DABC), Rua Profa. Zulmira Canavarros, 95, Centro, Cuiabá 78005-200, Brazil; dlflumig@yahoo.com.br; 5Institute of Chemistry, Center for Monitoring and Research of the Quality of Fuels, Biofuels, Crude Oil and Derivatives—CEMPEQC, São Paulo State University (UNESP), Araraquara 14800-900, Brazil; leticiaksantos@yahoo.com; 6Centro Universitário SENAI-CIMATEC, Av. Orlando Gomes, 1845, Piatã, Salvador 41650-010, Brazil; alexandre.wentz@fieb.org.br; 7Laboratório de Ensaios de Combustíveis, Universidade Federal de Minas Gerais, Avenida Antônio Carlos, 6627-Belo Horizonte, Belo Horizonte 31270-901, Brazil; vmdpasa@gmail.com (V.M.D.P.); regiane.debora@yahoo.com.br (R.D.F.R.)

**Keywords:** hydrophilic sulfonated silica catalyst, Aliquat 336, tetrabutylammonium tetrafluoroborate, transesterification, fatty acid methyl esters

## Abstract

SiO_2_-SO_3_H, with a surface area of 115 m^2^·g^−1^, pore volumes of 0.38 cm^3^·g^−1^ and 1.32 mmol H^+^/g, was used as a transesterification catalyst. Triglycerides of waste cooking oil reacted with methanol in refluxing toluene to yield mixtures of diglycerides, monoglycerides and fatty acid methyl esters (FAMEs) in the presence of 20% (*w*/*w*) catalyst/oil using the hydrophilic sulfonated silica (SiO_2_-SO_3_H) catalyst alone or with the addition of 10% (*w*/*w*) co-catalyst/oil [(Bu^n^_4_N)(BF_4_) or Aliquat 336]. The addition of the ammonium salts to the catalyst lead to a decrease in the amounts of diglycerides in the products, but the concentrations of monoglycerides increased. Mixtures of (Bu^n^_4_N)(BF_4_)/catalyst were superior to catalyst alone or Aliquat 336/catalyst for promoting the production of mixtures with high concentrations of FAMEs. The same experiments were repeated using DMSO as the solvent. The use of the more polar solvent resulted in excellent conversion of the triglycerides to FAME esters with all three-catalyst media. A simplified mechanism is presented to account for the experimental results.

## 1. Introduction

According to the definition provided by the American Society for Testing and Materials (ASTM), fatty acid methyl(ethyl) esters (biodiesel) is a fuel consisting of “long chain fatty acids of mono-alkyl esters derived from renewable fatty raw material such as animal fats or vegetable oils” [1]. In the search for an environmentally friendly method for biodiesel synthesis by transesterification of triglycerides, several alternatives for catalysis have been explored. In general, catalysts that can be used for producing biodiesel are divided into three categories: acidic, alkaline and biocatalysts. Acidic and alkaline catalysts are classified into two groups: homogeneous and heterogeneous catalysts.

Catalysts play a vital role in the transesterification process. Both the amount and type of catalyst affect the rate of reaction and conversion efficiency. Homogeneous catalysts function in the same phase as the reactants and can be categorized into homogeneous base catalysts and homogeneous acid catalysts. Actually, most fatty acid methyl esters (FAME) are produced by the base-catalyzed transesterification reaction due to its high conversion rate, negligible side reactions, and short reaction time. It is a low-pressure and low-temperature process, which occurs without the formation of intermediate substances. Despite these advantages, homogeneous base catalysts have some weaknesses. The production of biodiesel from feedstocks with a high free fatty acid (FFA) content is limited. It was reported by some researchers that homogeneous base catalysts are only effective for the FAME production via the transesterification process using the feedstocks with an FFA content of less than 2 wt% [2]. When FFA content is >2%, the catalyst reacts with FFA to produce soap and water. The soap inhibits the separation of FAME and glycerin, and the water can hydrolyze the esters in a reaction that competes with the transesterification.

Hydrochloric acid (HCl) and sulphuric acid (H_2_SO_4_) are generally used as acid catalysts, particularly when the vegetable oil contains a large amount of water and FFA. Waste cooking oil (WCO) is converted into FAME using these catalysts because these feed stocks have high concentrations of FFAs. The rate of reaction of a homogeneous acid catalyst is lower than that of a homogeneous base catalyst. These reactions require an excess of alcohol and a high temperature. Environmental problems and corrosion are also caused by liquid acids. The rate of the alkali-catalyzed transesterification reaction is nearly 4000 times faster than that of the acid-catalyzed transesterification reaction [3].

In transesterification reactions catalyzed by homogeneous bases or acids, salts of the base and glycerol are produced as by-products. Glycerol is usually used in the pharmaceutical industry after purification, and the excess KOH catalyst is used for the production of potassium fertilizer by neutralization with H_3_PO_4_ [4].

In recent years, a new approach for using heterogeneous catalysts to produce biodiesel has attracted considerable attention. Heterogeneous catalysts can be categorized into heterogeneous base catalysts and heterogeneous acid catalysts. For the transesterification processes, a number of heterogeneous (solid) catalysts have been tested at the laboratory scale. A recent review discusses the use of heteropoly acids (HPAs) and polyoxometalate compounds [5]. The flexibility of their structures and super acidic properties can be enhanced by incorporation of polyoxometalate anions into the complex proton acids. This pseudo liquid phase makes it possible for nearly all mobile protons to take part in the catalytic process. The rate of reaction is higher when homogeneous base catalysts are used, but the cost of the process is high because of the need to separate the catalysts from the reaction media after product formation. In another study, Encinar et al. [6] studied the use of NaNO_3_/SiAl as a heterogeneous catalyst for FAME production through transesterification of rapeseed oil with methanol. The highest yields of biodiesel containing 99% FAME were obtained with 5% *w*/*w* catalyst, with stirring at 700 rpm, with a 9:1 methanol/oil ratio at 65 °C. Hu et al. [7] carbonized ginger straw and sulfonated the amorphous carbon product to obtain an acidic catalyst with a -SO_3_H group density of 1.05 mmol/g. The catalyst was thermally stable up to 200 °C, and its use resulted in 93.2% conversion during the esterification of oleic acid using methanol (9:1 M ratio of methanol to oleic acid; catalyst loading, 7 wt%; reaction temperature, 64 °C; reaction time, 210 min). The catalytic activity decreased with recycling.

Gonçalves et al. [8] utilized a heterogeneous magnetic acid catalyst MoO_3_/SrFe_2_O_4_ composed of molybdenum oxide (MoO_3_) supported on strontium ferrite (SrFe_2_O_4_) to catalyze the transesterification of waste cooking oil. They obtained 95.4% conversion into esters at 164 °C using a 40:1 molar ratio of alcohol-to-oil, 10% catalyst and a reaction time of 4 h. The catalyst was recovered by the use of an external conventional magnet. The ester content remained above 84% after five runs.

In another review [9], the conversion and utilization of agricultural wastes as heterogeneous catalysts for biodiesel production, as well as the types of catalysts and the effects of conversion and modification techniques on pore size, acidity, and surface area, are discussed. Mansir et al. [10] also discussed the advantages of the use of heterogeneous catalysts over that of homogeneous catalysts for the conversion of low-grade oils to biodiesel. The presence of free fatty acids in the oil interferes with homogeneous catalysis by KOH or NaOH because the fatty acids neutralize the bases to form soap, which interferes with the separation of the FAME and the glycerol in the product. The hydrolysis of the esters by the water formed also interferes with the transesterification reaction. In addition, the major drawbacks of basic heterogeneous catalysts are soap formation, leaching of catalytic active sites in reaction medium during reaction, and high sensitivity to moisture and FFA (>1%). Low stability during storage in the presence of water and CO_2m_ is another disadvantage [11]. There is a need for acid pre-treatment of such low-quality feedstocks to reduce FFA content to a bearable amount prior to base-catalytic transesterification to use high FFA feedstocks in a single-run process [12]. These limitations in the use of heterogeneous basic catalysts for biodiesel production are the principal driving forces for the development of new catalytic systems for biodiesel synthesis that are easier to manage and reuse. Previous research showed that some heterogeneous acidic catalysts were highly-active and recyclable with little deactivation during the conversion of vegetable oils to biodiesel [12,13,14]. Pore diameter is another factor that influences the accessibility of the oil to the catalytic site. Even in the case of acidic heterogeneous catalysts, leaching can limit the number of times that they can be recycled. There are other disadvantages in using heterogeneous catalysts. A larger amount of catalyst is required, as well as a higher temperature. The reaction rate is usually lower, and there can be a limitation of mass transfer. Deactivation due to the presence of impurities can occur, in addition to leaching. Side reactions such as ether formation can occur. Some of these disadvantages might be remedied by the use of a catalyst with a greater affinity for water, the use of a low oil to methanol ratio, the use of a support with larger pore sizes and a larger specific surface area or the use of co-solvents [10].

Another review [15] also discusses the preparation of heterogeneous catalysts from biomass. The fact that biomass is abundantly available, a cheaper raw material, reusable, non-toxic and biodegradable is considered to be an advantage over commercially available catalysts. Carbon material synthesized from biomass acts as the efficient support for active sites because of its high porosity and surface area. Biomass-derived basic, acidic and magnetic heterogeneous catalysts have been developed through several pathways that begin with the synthesis of the Appendix A (carbon) and terminate with the functionalization process to form the catalyst. Sulfonation with sulfuric acid and sulfonation by reduction and arylation have been used. Most biomass-based catalysts furnished greater than 80% yields of biodiesel under optimum conditions.

In an exception to this tendency to use acidic heterogeneous catalysts, a review by Jayakumar et al. [16] discusses the preparation of basic heterogeneous catalysts by the calcination of animal shells and bones. Over 98% yields could be obtained in the transesterification of fats and oils employing these catalysts. An interesting alternative is reviewed by Mulyatun et al. [17], where a combination of acid-base bifunctional heterogeneous catalysis is employed. Low grade feedstocks containing free fatty acids were converted to biodiesel. Mixed metal oxide catalysts were prepared and utilized for a combination of esterification and transesterification reactions involving Lewis bases and Bronsted and Lewis acids.

The development of solid acid catalysts has been studied by this group [18]. In principle, solid catalytic mixtures with large pores through which bulky triglycerides can access as many acidic sites as possible were sought. These sites should be highly stable, and large pores should also allow esterification of free fatty acids that are also present in vegetable oils [19]. Shah et al. [20] studied the kinetics of the conversion of waste cooking oil to biodiesel catalyzed by silica-sulfuric acid. Recently, the propyl sulfonic acid-functionalized silica, SiO_2_-Pr-SO_3_H, was synthesized from commercial silica gel and 3-mercaptopropyltriethoxysilane, followed by the oxidation of SiO_2_-Pr-SH to SiO_2_-Pr-SO_3_H with H_2_O_2_. The functionalized silica was applied as an alternative to traditional sulfuric acid or sulfonic resins for catalyzing chemical transformations [21,22,23,24,25]. Although it was employed for the preparation of biodiesel by the methanolic esterification of free fatty acids (FFAs) present in vegetable oils [20], in simulated oils [26], and in beef tallow [27], there is no report of the use of sulfuric acid-functionalized silicas for the production of biodiesel from triacylglycerides under atmospheric pressure.

As part of an ongoing study of the use of SiO_2_-SO_3_H for clean synthesis [28,29,30,31], we report herein the direct preparation of fatty acid methyl esters (FAMEs) using methanol, waste cooking oil, and the hydrophilic SiO_2_-SO_3_H as a catalyst, alone or in combination with the quaternary ammonium salts Aliquat 336 (ionic liquid) or (Bu^n^_4_N)(BF_4_) in refluxing toluene or in DMSO (Figure 1).

## 2. Experimental

All the reactions were performed in air under atmospheric pressure and monitored by thin layer chromatography (TLC) using pre-prepared plates (Silica Gel 60 F 254 on aluminum).

### 2.1. Raw Materials and Chemicals

Waste cooking oil (soybean) was donated by the university restaurant, and it was filtered through silica gel, which removed any polar and polymeric substances and fatty acids, prior to use. The physical parameters determined for the yellow oil were the viscosity (41.2 m·Pa) and the density (0.883 g·mL^−1^). All the other reagents (analytical grade), including dry toluene, DMSO and methanol, were supplied by Vetec, São Paulo, Brazil.

### 2.2. Preparation of the Silica Gel and Sulfonated Silica (SiO_2_-SO_3_H)

The preparation of silica gel and the sulfonated silica SiO_2_-SO_3_H catalyst have been reported previously [28,29,30,31].

### 2.3. Typical Procedures

#### 2.3.1. Reacting the Triglycerides from Waste Cooking Oil with Methanol Using the SiO_2_-SO_3_H Catalyst

The procedure utilized for the transesterification reaction was based on various trials to determine the optimum conditions for this reaction. Waste cooking oil (11.2520 g; 12.8856 mmol), methanol (22.50 mL, 17.8431 g, 556.90 mmol; or a molar ratio of waste cooking oil-to-methanol of 1:43), toluene (244.0 mL) and the catalyst SiO_2_-SO_3_H (2.2504 g; 20% *w*/*w* of waste cooking oil) were mixed in a 500-mL round bottom flask equipped with a reflux condenser, and the mixture was refluxed for 72 h at 110 °C. The mixture was cooled, and the solid catalyst was filtered. The methanol and toluene were evaporated separately on a rotary evaporator, purified by distillation and used in new reaction processes within this study. The organic layer was decanted into a separatory funnel, where the biofuel-containing upper phase was separated from the lower phase containing glycerol by decantation. The recovered glycerol was treated as previously described [4]. The biofuel phase was dissolved in hexane (50 mL), extracted with 20 mL of a saturated solution of NaCl, dried with MgSO_4_ and concentrated.

The same procedure was repeated using DMSO as the solvent with heating at 110 °C for 72 h. After cooling, the solid catalyst was filtered, and the excess methanol and DMSO were removed on a rotary evaporator under high vacuum. The glycerol was separated from the oil with a separatory funnel. The biodiesel was diluted in hexane, and the mixture was treated as described above. The biodiesel was finally dried under high vacuum for 5 h.

The catalyst was transferred to a muffle furnace, heated for 2 h at 200 °C, cooled and stored in a desiccator before reuse. The catalyst could be used a second time with yields similar to those of the first process. A third reaction using the same catalyst resulted in a 10% decrease in yield. The glycerol was purified by adsorption using activated charcoal, and it was used in the synthesis of ketals [4].

#### 2.3.2. Reacting the Triglycerides from Waste Cooking Oil with Methanol with the SiO_2_-SO_3_H Catalyst and (Bu^n^_4_N)(BF_4_)

Waste cooking oil (11.2520 g; 12.8856 mmol), methanol (22.50 mL, 17.8431 g, 556.90 mmol), toluene (244.0 mL), (Bu^n^_4_N)(BF_4_) co-catalyst (1.1252 g or 10% *w*/*w* of waste cooking oil) and the catalyst SiO_2_-SO_3_H (2.2504 g, 20% *w*/*w* of waste cooking oil) were mixed in a 500-mL round bottom flask equipped with a reflux condenser and refluxed at 110 °C for 72 h. The mixture was cooled, and the solid catalyst was filtered. The methanol and toluene were eliminated on a rotary evaporator, and the organic layer was decanted into a separatory funnel, where the upper phase containing biofuel was separated from the lower phase containing glycerol and (Bu^n^_4_N)(BF_4_) by decantation. The FAME phase was dissolved in hexane (50 mL) and extracted with 20.0 mL of a saturated solution of NaCl, dried with MgSO_4_ and concentrated.

The catalyst was transferred to a muffle furnace and heated for 2 h at 200 °C, cooled and stored in a desiccator before reuse. The glycerol was purified and separated from the (Bu^n^_4_N)(BF_4_) after dilution in 20 mL of methanol and filtration through a column containing silica gel. The methanol was eliminated by evaporation on a rotary evaporator. The (Bu^n^_4_N)(BF_4_) was extracted on a silica column and eluted with hexane (30 mL); the hexane was evaporated to recover the (Bu^n^_4_N)(BF_4_). This procedure was repeated using DMSO as the solvent, and the recovery of the biodiesel was performed as described above.

#### 2.3.3. Reacting the Triglycerides from Waste Cooking Oil with Methanol with SiO_2_-SO_3_H Catalyst and Aliquat 336

In a similar apparatus, waste cooking oil (11.2520 g; 12.8856 mmol), methanol (22.50 mL, 17.8431 g, 556.90 mmol), toluene (244.0 mL), Aliquat 336 co-catalyst (1.1252 g; 10% *w*/*w* of waste cooking oil) and the catalyst SiO_2_-SO_3_H (2.5040 g, 20% *w*/*w* of waste cooking oil) were mixed in a 500 mL round bottom flask equipped with a reflux condenser and refluxed at 110 °C for 72 h. The mixture was then cooled, and the solid catalyst was filtered. The methanol and toluene were evaporated on a rotary evaporator, and the organic layer was decanted into a separatory funnel, where the upper phase containing FAME was separated from the lower phase containing glycerol and Aliquat by decantation. The biofuel phase was dissolved in hexane (50 mL) and extracted with 20.0 mL of a saturated solution of NaCl, dried with MgSO_4_ and concentrated.

The catalyst was transferred to a muffle furnace, heated for 2 h at 200 °C, cooled and stored in a desiccator before reuse. The glycerol was purified and separated from the Aliquat after dilution in 20 mL of methanol and filtration through a column containing silica gel. The methanol was eliminated on a rotary evaporator. The Aliquat was eluted from a silica column by hexane (30 mL), and the hexane was evaporated to recover the Aliquat. This procedure was repeated using DMSO as the solvent, and the recovery of the biodiesel was achieved as described above.

### 2.4. Waste Cooking Oil and Biodiesel Analysis

The official methods proposed by ISO 12,966 were used to determine the compositional profile by gas chromatography with a flame ionization detector (GC-FID) (Shimadzu GC-2010). The chromatographic system used to separate and identify FFAs (wt%) included a cross-bound polyethyleneglycol capillary column (Supelco SP 2560, 100 m × 0.25 mm × 20 µm). The initial temperature was 60 °C for 2 min; the temperature increased to 220 °C at 10 °C·min^−1^, and finally, to 240 °C at 5 °C·min^−1^, where it was held for 7 min. The injector and detector temperatures were 350 °C, and the sample (0.5 µL injected) was dissolved in 99% iso-octane.

The methods of the European Standards (EN 14103) and the Brazilian Technical Standards Association (ABNT NBR 15908) were used to quantify FAMEs and remaining mono, di and triglycerides (MG, DG, TG) in the biodiesel. For quantification of FAMEs, a Thermo Trace GC-Ultra chromatograph was used, equipped with a flame ionization detector and a Thermo Scientific TR-BD (FAME) Capillary GC Column (L × I.D. 30 m × 0.25 mm, df 0.25 μm) containing a polyethyleneglycol stationary phase, according to the EN 14,103 analytical procedure. Pure methyl nonadecanoate (C19:0, Sigma-Aldrich, Cotia, Brazil) was used as an internal standard to normalize the peak areas of the chromatograms. The integration was achieved from the methyl hexanoate (C6:0) peak to that of the methyl nervonate (C24:1), including all the peaks identified as fatty acid methyl esters. To analyze the FAME samples, approximately 100 mg (accuracy ± 0.1 mg) of homogenized sample and approximately 100 mg (accuracy ± 0.1 mg) of nonadecanoic acid methyl ester were weighed in a 10 mL vial and diluted with 10 mL of toluene before the injection into the equipment. All the samples were prepared in duplicate. Chromatographic conditions are described as follows: (a) column temperature: 60 °C held for 2 min, programmed at 10 °C·min^−1^ to 200 °C, then programmed at 5 °C·min^−1^ to 240 °C; the final temperature was held for 7 min; (b) injector and detector temperature: 250 °C; (c) helium carrier gas flow rate: 1–2 mL·min^−1^, a minimum flow rate of 1 mL·min^−1^ was warranted when operating at the maximum temperature; (d) injected volume: 1 μL and (e) split flow: 100 mL·min^−1^.

For the quantification of the glycerides (MG, DG, and TG), a Shimadzu GC2010 chromatograph equipped with a flame ionization detector was used according to the ASTM D6584 analytical procedure. The chromatographic system was configured to separate and identify MG, DG and TG with a CrossbondTM 5% Phenyl/95% polydimethylsiloxane capillary column (Zebron ZB-5HT, 30 m × 0.32 mm × 0.1 mm—Phenomenex, Torrence, CA, USA) with on-column injection. The initial temperature in the capillary column was 50 °C (1 min); the temperature increased to 180 °C at 15 °C·min^−^^1^, to 230 °C at 7 °C·min^−1^, and finally to 380 °C at 20 °C·min^−1^, where it was held for 10 min. The injector and detector temperatures were 380 °C, and the sample (0.5 mL injected) was prepared using heptane 99%. ^1^H- and ^13^C-NMR spectra were recorded on Bruker *Avance* 400 and *Avance* 500 spectrometers. These data are included in the Appendix A.

## 3. Results and Discussion

The compositional profile analysis of the waste cooking oil used in this work is described in Table 1. The main fatty acids in that oil were linoleic (C18:2) and oleic acids (C18:1); accordingly, the mean molecular weight of the fatty acids was determined to be 277.41 g·mol^−1^, and the mean molecular mass of the triglycerides was 873.22 g·mol^−1^. This profile was considered for calculating the molar ratio of waste cooking oil-to-methanol for the transesterification reaction. The composition of the oil was very similar to that of soybean oil described in the literature [32].

### 3.1. Transesterification of Waste Cooking Oil Using Catalysis by SiO_2_-SO_3_H/Toluene, SiO_2_-SO_3_H/Aliquat 336/Toluene and SiO_2_-SO_3_H/(Bu^n^_4_N)(BF_4_)/Toluene

In the first phase of this study, a large excess of methanol dissolved in toluene was mixed with waste cooking oil and 20 wt% (based on the mass of waste cooking oil) of pure SiO_2_-SO_3_H, and the mixture was refluxed. On the basis of the TLC monitoring of the reaction, the total consumption of triglycerides occurred after 72 h. This time required to complete the reaction was probably the result of the high degree of hydrophilicity of the catalyst, SiO_2_-SO_3_H. This period was maintained for all subsequent reactions with the mixtures of 20 wt% SiO_2_-SO_3_H with 10 wt% Aliquat 336 or 10 wt% (Bu^n^_4_N)(BF_4_). The FAMEs and glycerides (MG, DG, and TG) contained in the biodiesel phase were confirmed by GC-FID using the methods defined in EN 14,103 and ASTM D6584. The wt% composition of the mixture (FAMEs and glycerides) in the products is presented in Figure 2 and Table 2; they represent the average values of five different measurements. After 72 h, only the reaction products obtained with SiO_2_-SO_3_H and (Bu^n^_4_N)(BF_4_) contained traces of unreacted triglycerides (Table 2). The choice of toluene as the solvent for the reactions and the use of quaternary ammonium salts as co-promoters were indicated by the recently discovered efficiency of the mixture toluene, (Bu^n^_4_N)(BF_4_) and SiO_2_-SO_3_H in the esterification of fatty acids with solketal [30]. The highest yield (64.2%) of FAMEs was obtained using the combination of (Bu^n^_4_N)(BF_4_) with SiO_2_-SO_3_H.

In our previous experiences with esterification reactions catalyzed by SiO_2_-SO_3_H [30,33], we pointed out the importance of the cationic intermediates [RC(OH)_2_]^+^, which should be formed by the protonation of the acid at the catalyst surface; this charged species would disperse in solution where it would react with its alcohol counterpart [18]. We also pointed out that the addition of a quaternary ammonium salt to that type of mixture could change the polarity of the toluene phase, stabilizing those cationic intermediates, and leading to very good yields of, for instance, linoleic acid solketal ester [30]. The mechanism of the formation of biodiesel (FAMEs) using the solid catalyst can be oversimplified by the assumption that the majority of the chemical transformations will occur at the surface of the catalyst. With this assumption in mind, the three stages for the complete conversion of TG into long-chain methyl esters should occur as depicted in Figure 1. The first stage, denoted by V1, would probably be the rate-determining step due to the difficulty in connecting the long apolar chains of the TG molecules to the polar catalyst surface because of the sheer size of the molecule, with the inherent degrees of steric hindrance. The molecule’s reactive sites would have to reach the catalyst’s reactive sites: in the present case, this encounter would produce a cationic intermediate such as depicted in Figure 3.

Once the diglyceride is produced with the release of the methyl ester, the alcohol functional group of the DG would have the necessary properties to anchor or, at least, to facilitate the approach of this new reagent to the active centers of the solid catalyst [18]. Therefore, we can assume with certain confidence that the velocity of the V2 (Figure 1) process is much faster than V1, which means that, once formed, the DG molecules trapped at or near the active centers of the catalyst surface would rapidly (in the time frame of the overall reaction) be converted into monoglycerides with the release of the second methyl ester. Within this reasoning, step V3 (Figure 1) would be almost as delicate as V1, also requiring the fine tuning of the reaction parameters in terms of time, temperature, solvent and co-catalyst. Because the DG molecule was anchored at the surface of the catalyst, the formation of the second alcohol functional group, due to the transformation DGs to MGs, should strengthen the forces binding that molecule to the highly hydrophobic catalyst surface. It is reasonable to assume, at this stage, that the formation of glycerol in the process V3 could possibly block the active center(s) where the actual process occurred because of the high affinity of the catalyst for that kind of polar, H-bond-prone molecule.

In this reaction scenario, the co-catalyst should play multiple tasks: lower the repulsion forces of the bulky TG molecules towards the catalyst surface to enhance the V1_init_ process, stabilize the cationic intermediates from V1–V3, and lower the attraction of glycerol to the active sites after completion of the V3 step, effectively freeing those sites for the constructive continuation of the reactions V1-V2-V3 sequences. The three systems under study are capable of cleaving triglycerides by transesterification reactions with methanol in refluxing toluene, but with slightly different behaviors, as is shown in the graph in Figure 2, constructed using the data from Table 2.

The samples obtained from the 72 h reactions catalyzed by SiO_2_-SO_3_H alone contained no TGs, but had higher percentages at 13.9% of DGs than the samples taken from the other reactions, a clear indication that the addition of quaternary ammonium salts are important to accelerate the transformation of DGs to MGs (step V2 in Figure 1). Interestingly, the reactions with the lone catalyst resulted in the lowest percentage of MGs (25.9%), which could indicate that the ammonium salts are effective in stabilizing cationic species derived from protonated MGs, which would slow down their conversion to FAME and glycerol (step V3 in Figure 1).

The appearance of 0.7% TG in the samples derived from the system that includes the (Bu^n^_4_N)(BF_4_) seems to suggest that this co-catalyst slightly inhibits the V1 process of the binding of TG to an active center of the solid catalyst and the formation of the DGs. This is either due to the poor interaction of the ammonium salt with the bulky TGs, by the strong stabilization of a cation such as that in Figure 3, thereby retarding the V1_final_ step, or both. However, the action of the (Bu^n^_4_N)(BF_4_) in these systems seems to be beneficial because only 2.5% of DGs were found in the samples of those reactions (Table 2). This fact indicates that the process, crudely described above as V2, is indeed very fast with this mixed catalyst. Aliquat 336 and the (Bu^n^_4_N)(BF_4_) were also tested separately as catalysts for this transesterification reaction without the presence of the SiO_2_-SO_3_H; however, no reaction was observed.

### 3.2. Transesterification of Waste Cooking Oil Using Catalysis by SiO_2_-SO_3_H/DMSO, SiO_2_-SO_3_H/Aliquat 336/DMSO and SiO_2_-SO_3_H/(Bu^n^_4_N)(BF_4_)/DMSO

In a recent study, DMSO was shown to be an efficient solvent for the SiO_2_-SO_3_H-catalyzed dehydration of fructose to 5-hydroxymethyl-2-furfural [31]. The interaction of DMSO with the catalyst led us to test the use of DMSO in transesterification reactions. Because of the inadequate results obtained using toluene with regard to the complete consumption of the di- and monoglycerides, the transesterification was performed using DMSO as the solvent in the second phase of the study. The remaining conditions for the reaction were the same as those used with toluene. The yields are presented in Table 3.

The yields of FAMEs obtained using DMSO as the solvent were significantly higher than those obtained with toluene with all the catalyst mixtures. The first possible explanation for this result is that the more polar DMSO would help to stabilize the protonated ester intermediate in the transesterification reaction. The second, less clear, possibility is that the more polar DMSO might facilitate the transport of the triglyceride and methanol to the catalyst surface. Finally, because the triglyceride is less soluble in DMSO than in toluene, there might be a greater tendency for it to be absorbed on the catalyst surface, creating a concentration effect. There was essentially no difference in the yields obtained using SiO_3_-SO_3_H (99.51%) and SiO_3_-SO_3_H/(Bu^n^_4_N)(BF_4_) (98.40%). The SiO_3_-SO_3_H/(Bu^n^_4_N)(BF_4_) mixture was effective for the transesterification of triacylglycerols in DMSO and the esterification of free fatty acids with solketal [30]. These values are within the limits established by EN 14103-2011 (Min. 96.5%).

With regard to the concentration of total glycerol, the SiO_3_-SO_3_H (0.08%) and SiO_3_-SO_3_H/(Bu^n^_4_N)(BF_4_) (0.23%) catalytic systems were within the limits specified by ASTM D6584-17 (Max. 0.25%), which also demonstrates the high degree of conversion of the tri-, di- and monoacylglycerols. The concentration of total glycerol obtained with the SiO_2_-SO_3_H/Aliquat 336 catalytic system (1.62%) indicates that this system is less adequate for catalyzing the transesterification of waste cooking oil.

The concentration of diglycerides in the product obtained using the SiO_3_-SO_3_H catalyst (0.07%) in DMSO was within the limits established by ASTM D6584-17 (Max. 0.20%), but those obtained using the SiO_2_-SO_3_H/Aliquat 336 (4.00%) and SiO_3_-SO_3_H/(Bu^n^_4_N)(BF_4_) (0.37%) systems were outside that limit. The concentration of monoglycerides observed using SiO_3_-SO_3_H (0.28%) as the catalyst was within the limit imposed by the ASTM D6584-17 (Max. 0.7%), whereas those obtained using the SiO_3_-SO_3_H/(Bu^n^_4_N)(BF_4_) (0.86%) and SiO_2_-SO_3_H/Aliquat 336 (4.87%) catalytic systems were greater than the legal limit.

The yields of FAME obtained by Encimar et al. [32] from soybean oil using homogeneous catalysts was 97.6%, which was similar to those obtained in this work using SiO_2_-SO_3_H/(Bu^n^_4_N)(BF_4_) or SiO_2_-SO_3_H alone as the catalyst. The viscosity of the biodiesel obtained using SiO_3_-SO_3_H/(Bu^n^_4_N)(BF_4_) as the catalyst was 4.44 cSt and the density was 0.867 g·mL^−1^. The biodiesel obtained from soybean oil by Encimar et al. [32] had a density of 0.8857 g·mL^−1^ and a kinematic viscosity (40 °C) of 4.04 cSt. Biodiesel (ASTM D 6751) has an allowable viscosity range of 1.9 to 6.0 mm^2^/s. EN 14,214 differs in the range of allowable viscosity. FAME that meets EN 14,214 must have a viscosity between 3.5 to 5.0 mm^2^/s.

Shah et al. [33] prepared a solid catalyst by treating silica gel with chlorosulfonic acid and varied the conditions for the catalyzed transesterification of waste cooking oil (Transesterification reactions were carried out in a 700-mL laboratory-stirred autoclave (Amar Equipments, India; equipped with pressure gauge, temperature controller, internal cooling coil and sample outlet valve). They studied the effects of concentration of catalyst, temperature, time and ratio of oil to methanol. The highest yield (98%) was obtained using 3% (*w*/*w*) of catalyst at 120 °C over a 6-h period and a 1:20 ratio of oil-to-methanol. The biodiesel produced had a specific gravity of 0.880, a kinematic viscosity (40 °C) of 5.6 mm·s^−1^, an acid point < 0.1, a cloud point of −2.8 °C, and a cetane number of 60.5. They also compared their results with the yields obtained by other authors using sulfuric acid (90%) [34], carbon-based solid acids (80.5%) [35], Al(HSO_4_)_3_ (81%) [36], carbohydrate-derived solid acids (93%) [37], sulfated tin oxide (92.3) [38], and SO_4_^2−^/TiO_2_–SiO_2_. (94%) [39]. These yields were all lower than those obtained in our study. In addition, the yield was achieved at atmospheric pressure in our work, whereas that of Shah et al. [33] was obtained in a closed system under pressure.

Turnover number (TON) is moles of desired product formed/number of active centers or surface area (heterogeneous catalyst) and turnover frequency, TOF = TON/time of reaction. The turnover number calculated for the reaction catalyzed by SiO_2_-SO_3_H was 5.61 mmol·g^−1^, and the TOF was 0.78 mmol·g^−1^·h^−1^. The turnover frequency (TOF) and number (TON) have been revisited with the objective of forming a common ground to compare different catalysts. As Laidler wrote [40]: “However, the turnover number varies with the temperature, the concentration of substrate … and other conditions. Therefore, it is not a useful quantity in kinetic work …”.

## 4. Conclusions

The sulfonated silica SiO_2_-SO_3_H and the mixtures of SiO_2_-SO_3_H with Aliquat 336 and with (Bu^n^_4_N)(BF_4_) were used successfully for the transesterification of waste cooking soy oil with methanol in refluxing toluene. Analysis of the quantities of triglycerides, diglycerides, monoglycerides and fatty acid methyl esters in the reaction’s products indicated different behaviors of the catalysts. Whereas the unmixed sulfonated silica was the fastest in promoting the transformation of monoglycerides to FAMEs, the mixed catalysts favored the transformation of diglycerides into monoglycerides. The salt (Bu^n^_4_N)(BF_4_) showed a high efficiency in producing mixtures with high percentages of monoglycerides and FAMEs, despite its difficulty in promoting the initial approach of the triglyceride to the sulfonated catalyst surface. When DMSO was used in place of toluene under the same reaction conditions, the catalytic systems tested were effective for the transformation of tri-, di- and monoacylglycerols into FAME and glycerol. The results obtained using the sulfonated silica, SiO_3_-SO_3_H, and the SiO_2_-SO_3_H/(Bu^n^_4_N)](BF_4_) in DMSO were within the standards established by the ASTM, indicating that these catalytic systems are effective for the transformation of low quality cooking oils.

## Figures and Tables

**Figure 1 molecules-27-00953-f001:**
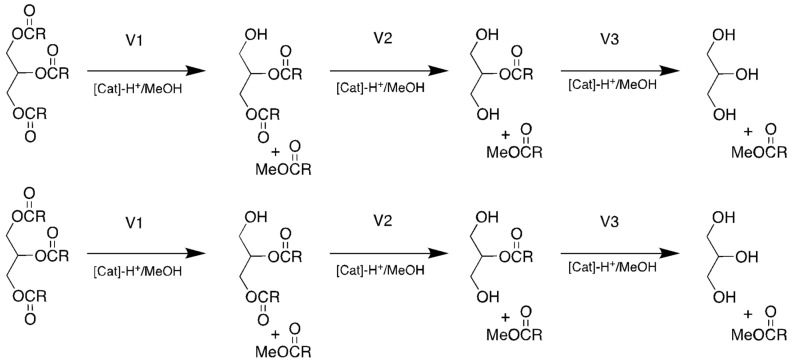
Schematic synthesis of FAME and glycerol (V1 = step one, V2 = step two and V3 = step three).

**Figure 2 molecules-27-00953-f002:**
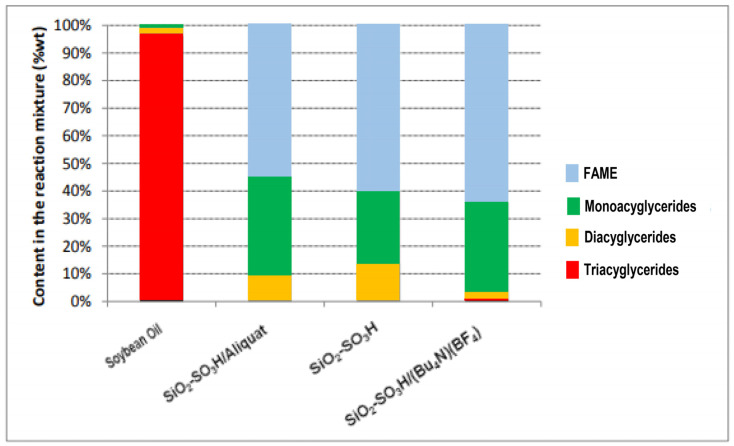
Composition of the waste cooking oil feedstock and the product mixture of each catalytic process (SiO_2_-SO_3_H/Aliquat, SiO_3_-SO_3_H and SiO_3_-SO_3_H/(Bu^n^_4_N)(BF_4_) involving methanol in refluxing toluene.

**Figure 3 molecules-27-00953-f003:**
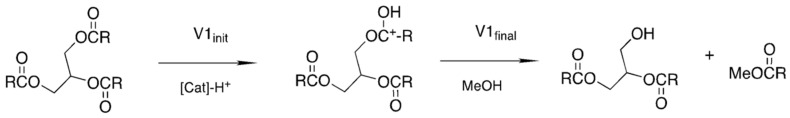
Formation of a diglyceride from the initial protonation of a triglyceride by the solid catalyst.

**Table 1 molecules-27-00953-t001:** Fatty acid composition of the waste cooking oil used in the present work.

Fatty Acid	Molecular Weight (g·mol^−1^)	wt%
Palmitic acid (C16:0)	256.43	10.41
Stearic acid (C18:0)	284.48	3.91
Oleic acid (C18:1)	282.46	26.52
Linoleic acid (C18:2)	280.45	51.66
Linolenic acid (C18:3)	278.43	5.55
Others	-	1.95
Mean Molecular Weight of Fatty Acids (g·mol^−1^)	277.41	
Mean molar mass of triglycerides (g·mol^−1^)	873.22	

Lit [32]: Palmitic acid, 11.6%; stearic acid, 3.22%; oleic acid, 25.09%; linoleic acid, 52.93%; linolenic acid, 5.95%; others, 1.08%.

**Table 2 molecules-27-00953-t002:** Composition of the waste cooking oil feedstock and the product mixture of each catalytic process (SiO_2_-SO_3_H/Aliquat, SiO_3_-SO_3_H and SiO_3_-SO_3_H/(Bu^n^_4_N)(BF_4_) involving methanol in refluxing toluene.

Products	Waste Cooking Oil	SiO_2_-SO_3_H/Aliquat	SiO_2_-SO_3_H	SiO_2_-SO_3_H/[(Bu^n^_4_N)](BF_4_)
Triacylglycerides (%)	96.2	0	0	0.7
Diacylglycerides (%)	2.80	9.5	13.9	2.5
Monoacylglycerides (%)	1	35.6	25.9	32.5
FAME (%)	0	54.9	60.2	64.2

Table 2. Numerical data for Figure 2.

**Table 3 molecules-27-00953-t003:** Composition of the waste cooking oil feedstock and the product mixture of each catalytic process SiO_3_-SO_3_H/(Bu^n^_4_N)(BF_4_), SiO_2_-SO_3_H/Aliquat 336, and SiO_3_-SO_3_H involving methanol in refluxing DMSO.

	SiO_2_-SO_3_H/(Bu^n^_4_N)(BF_4_)		SiO_2_-SO_3_H/Aliquat 336		SiO_2_-SO_3_H	
Assay	Result (% *w*/*w*)	Standard Deviation	Result (% *w*/*w*)	Standard Deviation	Result (% *w*/*w*)	Standard Deviation
FAME	98.40	3.10	86.35	3.38	99.51	2.94
Free glycerol	0.01	0,01	0.04	0.01	0.01	0.01
Total glycerol	0.29	0.07	2.16	0.08	0.09	0.07
Monoacylglycerol	0.86	0.27	4.87	0.29	0.28	0.26
Diacylglycerol	0.37	0.12	4.00	0.13	0.07	0.12
Triacylglycerol	0.09	0.26	0.35	0.28	0.02	0.25

## Data Availability

The data presented in this study are available on request from the corresponding author.

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
