# Peer review of "Catalytic Transformation of Triglycerides to Biodiesel with SiO2-SO3H and Quaternary Ammonium Salts in Toluene or DMSO"

_molecules, 2022, doi:10.3390/molecules27030953_

Round 1
Reviewer 1 Report
Dear authors,
Your article seems interesting, dealing with a subject that should always be encouraged, in order to improve the sustainability of biofuel production. In general, your article is focused in specific aspects of transesterification, which is good, as you have perfectly explained these aspects. However, your study lacks of some additional data in order to broaden the knowledge included in this article. Thus, I present the following concerns:
- Title: In my opinion, it is a bit long, you could shorten it a bit to make it more attractive to the readers. But this is a personal choice, so it is up to you (When I choose a title, I really like to keep it if possible).
- Keywords: If your article is finally published, you should avoid to repeat words that are included in the title, replacing it by other keywords, in order to increase the impact of your work in search engines.
- The introduction is too short. You need to extend it in order to give a proper background to the audience. For instance, you should talk about heterogeneous catalysis, its comparison (advantages and disadvantages) with homogeneous ones, the use of acid/base heterogeneous catalysts and different applied studies (paying attention to their effect on free fatty acid), and then you can put your work in context. In addition, there are very few references. Add the following (many of them reviews) to enrich your work (in addition, some results obtained in these articles can be useful for you, so take them into account for the results and discussion section):
- https://doi.org/10.3390/catal11091085
- https://doi.org/10.3390/catal11111405
- https://doi.org/10.1016/j.renene.2021.05.098
- https://doi.org/10.1016/j.fuel.2021.121463
- https://doi.org/10.1016/j.joei.2021.06.017
- http://dx.doi.org/10.1016/j.enconman.2016.07.037
- https://doi.org/10.1016/j.rser.2018.04.056
- https://doi.org/10.1016/j.biortech.2021.125054
- https://doi.org/10.1016/j.fuel.2021.122749
- Experimental: It seems more logical that subsection 2.1. be the last subsection in this part. Once you obtain your samples, you characterize them.
- Subsection 2.2. What is the nature of this oil? By the look of it, it seems to be soybean oil used in restaurants (according to the FA profile and the conclusion section). In any case, the majority oil seems to be soybean oil. You should indicate it, and add some basic characteristics of the oil (density, viscosity, % of impurities after filtration, etc.).
- Subsections 2.4.1, 2.4.2, and 2.4.3. The experimental conditions were based on previous studies? You should point it out. Instead of talking about mmol, you should talk about oil:methanol ratio. You talk about the possible reuse of the catalyst in these sections, preparing the sample for this purpose. How was this reusability? Did you try any reusability tests? This could be interesting, as it is one of the main challenges of heterogeneous catalysts (with only 2 or 3 times reused).
- Results and discussion: Line 175: Commercial brand? It seems a different oil, maybe you mean that the commercial brand was used and then you studied the subsequent waste cooking oil?
- Table 1. You should compare with other oils, to check if it is a typical soybean oil or some mixtures were found with other oils in the restaurant. For that purpose, compare with these two articles:
- https://doi.org/10.1016/j.foodchem.2020.127612
- http://dx.doi.org/10.1016/j.biombioe.2014.01.034
- In general, you should support your explanations in this section with the literature, apart from your previous works.
- For instance, in Table 3, you can compare with other methods included in the literature (in the previous references included in Table 1 and the introduction you can find many related results) to enhance the good results that you have obtained in this case. In addition, why not compare other aspects of the standard, as viscosity or density, which are easily measured and (according to the high yield obtained) will be within the standards? Then you can confirm that your biodiesel is suitable for other different aspects, and compare with other soybean biodiesel obtained in the literature, which will broaden the knowledge included in this work.
- References: As I told you previously, there are very few references. You should include more (apart from the ones included in this review) in order to discuss your results, to improve your introduction, etc. Apart from that, the references added should be recent, in order to prove that your article is still interesting for the scientific community. Many of the articles that I recommend you add are quite recent.
To sum up, your article has a lot of potential, but some improvements are required to increase the quality of your work.
Author Response
Title: In my opinion, it is a bit long, you could shorten it a bit to make it more attractive to the readers.
The title was shortened to
“Catalytic transformation of triglycerides to biodiesel with SiO2-SO3H and quaternary ammonium salts in toluene or DMSO”
Keywords: If your article is finally published, you should avoid to repeat words that are included in the title, replacing it by other keywords, in order to increase the impact of your work in search engines.
The keywords are no longer the same as the words in the title.
The introduction is too short. You need to extend it in order to give a proper background to the audience. For instance, you should talk about heterogeneous catalysis, its comparison (advantages and disadvantages) with homogeneous ones, the use of acid/base heterogeneous catalysts and different applied studies (paying attention to their effect on free fatty acid), and then you can put your work in context.
The introduction was extended with the inclusion of the references indicated by the reviewer.
Experimental: It seems more logical that subsection 2.1. be the last subsection in this part. Once you obtain your samples, you characterize them.
Section 2.1 was moved to the end of the experimental section, becoming section 2.4.
What is the nature of this oil? By the look of it, it seems to be soybean oil used in restaurants (according to the FA profile and the conclusion section). In any case, the majority oil seems to be soybean oil. You should indicate it, and add some basic characteristics of the oil (density, viscosity, % of impurities after filtration, etc.).
The nature of the oil as soybean oil was indicated in the text, as well as the density and viscosity of the oil and the method of purification by filtering through silica gel.
The experimental conditions were based on previous studies? You should point it out.
The experimental conditions were based on a series of preliminary experiments to establish the optimum conditions. This fact was added to the text.
Instead of talking about mmol, you should talk about oil:methanol ratio.
The molar ratio of oil to methanol was included in the text.
You talk about the possible reuse of the catalyst in these sections, preparing the sample for this purpose. How was this reusability? Did you try any reusability tests? This could be interesting, as it is one of the main challenges of heterogeneous catalysts (with only 2 or 3 times reused).
The catalyst was reused twice. The first time the yield was similar to the original yield. The second time, there was a decrease of 10% in the yield. This fact was added to the experimental section.
Results and discussion: Line 175: Commercial brand? It seems a different oil, maybe you mean that the commercial brand was used and then you studied the subsequent waste cooking oil?
The text was correct to “waste cooking oil”.
Table 1. You should compare with other oils, to check if it is a typical soybean oil or some mixtures were found with other oils in the restaurant. For that purpose, compare with these two articles:
The composition of the soybean oil used in the second article cited by the review was included under Table 1 for comparison. The first article de not list the composition of soybean oil, only the composition of poultry ration containing soybean oil.
- In general, you should support your explanations in this section with the literature, apart from your previous works. For instance, in Table 3, you can compare with other methods included in the literature (in the previous references included in Table 1 and the introduction you can find many related results) to enhance the good results that you have obtained in this case.
The results obtained in other studies have been included for comparison.
In addition, why not compare other aspects of the standard, as viscosity or density, which are easily measured and (according to the high yield obtained) will be within the standards?
The density and the viscosity of the product were cited and compared with values from the literature and with official specifications.
You should include more (apart from the ones included in this review) in order to discuss your results, to improve your introduction, etc.
Various other references were included in the text, including recent articles.
Reviewer 2 Report
The paper concerns transformation of triglycerides to fatty acid methyl esters with SiO2-SO3H and quaternary ammonium salts in toluene or DMSO. The results showed that the unmixed sulfonated silica was the fastest in promoting the transformation of monoglycerides to FAMEs, the mixed catalysts favored the transformation of diglycerides into monoglycerides. The salt (Bu4N)(BF4) showed a high efficiency in producing mixtures with high percentages of monoglycerides and FAMEs. When DMSO was used in place of toluene under the same reaction conditions, the catalytic systems tested were effective for the transformation of tri-, di- and monoacylglycerols in FAME and glycerol. So I agree to publish the paper after revision.
- In line 31, [(Bun4N)](BF4) or Aliquat 336] should be put after “10% (w/w) co-catalyst”. In line 59, is (Bu4N)(BF4) right? In figure 1, V1, V2 and V3 should be noted. In line 65, TLC should be noted.
- Why is reaction time 72 h so long? Why did not you change reaction condition such as temperature, time for the experiments to obtain best condition?
- [Bun4N)](BF4) or Aliquat 336 serving as co-catalyst, whether does alone [Bun4N)](BF4) or Aliquat 336 possess catalytic ability to the reaction?
Author Response
- In line 31, [(Bun4N)](BF4) or Aliquat 336] should be put after “10% (w/w) co-catalyst”. In line 59, is (Bu4N)(BF4) right? In figure 1, V1, V2 and V3 should be noted. In line 65, TLC should be noted.
- The corresponding adjustments were made to the text.
- Why is reaction time 72 h so long? Why did not you change reaction condition such as temperature, time for the experiments to obtain best condition?
The procedure used was based on various preliminary reactions to determine the optimum conditions for the reactions. The temperature used was that corresponding to the boiling point of the solvent mixture.
The long time necessary for the completion of the reaction was probably the result of the high degree of hydrophilicity of the catalyst. This fact is now mentioned in the text.
- [Bun4N)](BF4) or Aliquat 336 serving as co-catalyst, whether does alone [Bun4N)](BF4) or Aliquat 336 possess catalytic ability to the reaction
As was already mentioned in the text, the yield of biodiesel was 10% in the presence of the quaternary salts. This small degree of catalysis was probably the result of general acid-base catalysis.
Reviewer 3 Report
1) The authors have utilized waste cooking oil for biodiesel synthesis using green approach.
2) The catalyst is green but synthesis of biodiesel requires longer reaction time with harsh reaction condtions.
Catalyst characterization is not provided in the manuscript.
TON and TOF number should be added in to the manuscript.
Add comparative study with reported methods.
Author Response
1) The authors have utilized waste cooking oil for biodiesel synthesis using green approach.
2) The catalyst is green but synthesis of biodiesel requires longer reaction time with harsh reaction condtions.
Catalyst characterization is not provided in the manuscript.
The characterization of the catalyst has been previously reported, and the repetition of the description in this work was considered to be redundant.
TON and TOF number should be added in to the manuscript.
The following text was added to the manuscript:
The turnover number calculated for the reaction catalyzed by SiO2-SO3H was 5.61 mmol.g-1, and the TOF was 0.78 mmol.g-1.h-1. The turnover frequency (TOF) and number (TON) have been revisited with the objective of forming a common ground to compare different catalysts. As Laidler wrote [40] “However, the turnover number varies with the temperature, the concentration of substrate... and other conditions. Therefore it is not a useful quantity in kinetic work...”
Add comparative study with reported methods.
The results obtained in other studies have been included for comparison.
Round 2
Reviewer 2 Report
Authors have revised some errors and I agree to its publication.